# Stepwise Corrected Attention Registration Network for Preoperative and Follow-Up Magnetic Resonance Imaging of Glioma Patients

**DOI:** 10.3390/bioengineering11090951

**Published:** 2024-09-23

**Authors:** Yuefei Feng, Yao Zheng, Dong Huang, Jie Wei, Tianci Liu, Yinyan Wang, Yang Liu

**Affiliations:** 1School of Biomedical Engineering, Air Force Medical University, No. 169 Changle West Road, Xi’an 710032, China; deer_von@163.com (Y.F.); zhengyao0202@fmmu.edu.cn (Y.Z.); huangdong1007785@outlook.com (D.H.); jieweiwtt@fmmu.edu.cn (J.W.); liutianci@fmmu.edu.cn (T.L.); 2Shaanxi Provincial Key Laboratory of Bioelectromagnetic Detection and Intelligent Perception, No. 169 Changle West Road, Xi’an 710032, China; 3Department of Neurosurgery, Beijing Tiantan Hospital, Capital Medical University, No. 119 Area A, Nansihuanxi Road, Beijing 100070, China

**Keywords:** deformable registration, multi-level registration, non-correspondence, unsupervised learning, corrected attention

## Abstract

The registration of preoperative and follow-up brain MRI, which is crucial in illustrating patients’ responses to treatments and providing guidance for postoperative therapy, presents significant challenges. These challenges stem from the considerable deformation of brain tissue and the areas of non-correspondence due to surgical intervention and postoperative changes. We propose a stepwise corrected attention registration network grounded in convolutional neural networks (CNNs). This methodology leverages preoperative and follow-up MRI scans as fixed images and moving images, respectively, and employs a multi-level registration strategy that establishes a precise and holistic correspondence between images, from coarse to fine. Furthermore, our model introduces a corrected attention module into the multi-level registration network that can generate an attention map at the local level through the deformation fields of the upper-level registration network and pathological areas of preoperative images segmented by a mature algorithm in BraTS, serving to strengthen the registration accuracy of non-correspondence areas. A comparison between our scheme and the leading approach identified in the MICCAI’s BraTS-Reg challenge indicates a 7.5% enhancement in the target registration error (TRE) metric and improved visualization of non-correspondence areas. These results illustrate the better performance of our stepwise corrected attention registration network in not only enhancing the registration accuracy but also achieving a more logical representation of non-correspondence areas. Thus, this work contributes significantly to the optimization of the registration of brain MRI between preoperative and follow-up scans.

## 1. Introduction

Deformable medical image registration is a critical and fundamental task in various clinical applications, such as preoperative surgery planning, image-guided intervention, the monitoring of patients’ responses to treatments, and postoperative therapy [1,2,3,4,5,6,7,8]. It aims to establish a spatial correspondence between medical images from different times, different patients, or different devices by searching for and computing dense and nonlinear deformation fields. The accurate registration of preoperative and follow-up images is pivotal in the whole medical management process and especially in the individual-based treatment of glioma patients. These images serve as critical guides for the assessment of the impact of treatments such as surgical resection and for planning further therapeutic strategies. However, the registration process is fraught with challenges due to the temporally large deformation of brain tissue [9] and the large topological changes caused by disease progression in the brain [10].

The current registration methods have difficulties in handling registration between paired MRI scans with high variability and complexity in deformation. In particular, brain tissue becomes increasingly complex in response to surgery, which causes large and complex deformations. Moreover, the presence of resection cavities and postoperative changes in brain tissue can lead to significant discrepancies in corresponding anatomic structures, thereby complicating the task of aligning images accurately [11].

Many traditional methods have been proposed in the field of medical image registration over the past few decades, providing solutions to align mismatching medical data and thereby creating possibilities for the longitudinal analysis of disease progression, the evaluation of treatment effects, and other diagnostic procedures. These classical registration approaches, such as elastic matching, the demons approach, the B-spline method, and the SyN method, have achieved remarkable performance in various registration tasks [12,13,14,15]. The fundamental principle of these methods is to iteratively minimize a cost function measuring the discrepancy between image pairs, which is usually a high-dimensional mathematical optimization problem. However, while these methods have good registration performance, they also lead to high complexity and high computational costs.

Recent advancements in deep learning have sparked developments in registration methods, demonstrating notable performance enhancements over traditional algorithmic approaches. Supervised methods were the first to be introduced for the task of image registration [16], where a ground-truth deformation field is necessary for training. However, acquiring ground-truth deformation fields for medical images poses a significant challenge since they require expert annotations, which are labor-intensive, costly, and sometimes impractical. Consequently, many supervised methods resort to using synthetically generated deformations via traditional registration methods as labels for training [17,18]. While this approach provides a means to train models, it limits their registration performance to the accuracy of the traditional methods that are used to generate the labels.

Unsupervised learning approaches, which do not require ground-truth deformation fields for training, instead leverage the inherent features of images to deduce the optimal transformation [19,20,21]. These approaches utilize loss functions that capture the similarity between moving and fixed images, such as mutual information or normalized cross-correlation values, combined with regularization terms that ensure smoothness constraints on the deformation fields. Unsupervised registration methods have the advantage of being more adaptive across different datasets, as they are not confined to the specific deformations within a single training set of a single organ. They can discover plausible deformations, making them especially useful for medical images characterized by diverse and complex pathologies [22]. Moving forward, the integration of unsupervised deep learning strategies is poised to significantly impact the field of medical image registration, offering a powerful tool for the analysis and interpretation of medical images.

Multi-resolution registration approaches are regarded as some of the most sophisticated and effective methods for medical image registration, particularly due to their ability to reconcile the inherent trade-offs between the global and local registration accuracy. By employing multi-resolution image pyramids, these methods harness a systematic coarse-to-fine strategy [10,23,24], where lower-resolution images provide a macroscopic perspective for global alignment, while successive, higher-resolution levels incrementally refine the registration to capture more localized deformations. In practical application, these approaches have been shown to more efficiently achieve the searching and estimation of deformation fields, primarily because the multi-stage process reduces the dimensionality of the problem space at each level, allowing for more rapid convergence. Nevertheless, despite the clear strengths of multi-resolution strategies, they are not without limitations. A notable concern arises with the downsampling methods used to construct the image pyramids, which can lead to a loss of texture and fine details—attributes of the image that are crucial for accurate registration. While wavelet transformations have been employed to address this issue, offering an alternative that preserves more image information through multi-frequency decomposition [25], they do not always work. In particular, wavelet transformations cannot entirely compensate for challenges such as non-correspondence areas within the image pairs that do not have a direct counterpart due to surgical intervention or pathological changes, which are pervasive in preoperative and follow-up brain scans. Moreover, temporal changes in brain tissue texture and structure further compound the difficulty as no method can fabricate information that has been completely lost.

In this work, we propose a stepwise corrected attention method for registration between preoperative and follow-up MRI scans of glioma patients. Our approach takes advantage of a multi-level registration strategy that can perform registration from coarse images to fine images. Our approach allows our model to capture complex and large deformations with high efficiency and accuracy. A corrected attention module is also introduced into the stepwise registration network so that the model can continuously focus the attention on learning the deformation of large deformation areas that may be non-correspondence areas with a high probability. Our model has been validated on a public dataset and compared with state-of-the-art deformable medical image registration approaches. The main contributions of this work are summarized as follows:Our method is designed as a bidirectional registration framework to address the problem of non-correspondence voxels, avoiding the irrationality of the registration results of areas with missing correspondence and balancing the information input into the model from paired registered images.Our model embraces three levels of stepwise registration to capture the most accurate deformation field at the initial level and refines the deformation field based on the item output from the upper-level registration network at the following level. Thus, it is capable of generating an accurate and reasonable deformation field.To further eliminate adverse effects of possible registration biases, our model incorporates a corrected attention module that enhances the model’s focus on areas with significant deformation and integrates the clinical data of the area if that certain pathological area of the preoperative image should have no corresponding relationship.

## 2. Methods

### 2.1. Problem Statement

Traditional deformable image registration (DIR) aims to find a spatial correspondence between two or more images that capture the complex anatomical transformations of biological tissue. In particular, it satisfies this requirement by registering moving images $I_{m}\in R^{L\times W\times D}$ to fixed images $I_{f}\in R^{L\times W\times D}$. The calculation process can be expressed via the following optimization problem:(1)$$\phi^{*}=argmin\ell \left(I_{f},I_{m}\circ \phi \right)+\lambda R\left(\phi \right)$$
where $\phi \in R^{L\times W\times D\times 3}$ represents the deformation field that reveals the mapping relationship of voxels between the moving image $I_{m}$ and fixed image $I_{f}$. The term $I_{m}\circ \phi$ represents the warped image $I_{w}$ obtained by applying ϕ on $I_{m}$. The loss function *ℓ* is used to measure the difference between $I_{f}$ and $I_{w}$, and the other term R(ϕ) with a hyperparameter λ is used to regularize the influences of the optimization. However, preoperative and follow-up images of glioma patients often contain voxels lacking corresponding relationships. In particular, as shown in Figure 1, there are no voxels in follow-up image corresponding to the tumor tissue of the preoperative image due to surgical resection, and there are no voxels in the preoperative image corresponding to areas where cerebrospinal fluid is present in the follow-up image. Thus, traditional registration methods may lose efficacy, given their assumption of consistent representation across the images being registered. Therefore, we propose our stepwise corrected attention registration network.

### 2.2. Bidirectional Registration Framework

To address the problems mentioned in the above section, our approach utilizes a bidirectional registration framework with the intention of addressing the unique challenges associated with the registration of preoperative and follow-up medical images, as shown in Figure 2. Our bidirectional framework treats each image as a fixed image or moving image. Our bidirectional approach does not presume the existence of completely equivalent anatomical structures or features in pairs of registered images. Instead, it acknowledges that some reciprocal correspondences are non-existent. Inspired by Tony’s work [26], we utilize a forward–backward consistency constraint to calculate areas of non-correspondence. Specifically, the forward–backward (inverse consistency) error δ from preoperative to follow-up is defined as
(2)$$\delta \left(I_{f}\right)={∥\phi_{one}\left(I_{m}\right)+\phi_{other}\left(I_{m}+\phi_{one}\left(I_{m}\right)\right)∥}_{2}$$
where $\phi_{one}$ and $\phi_{other}$ represent the deformation field in the forward and backward directions, respectively. We estimate the areas of non-correspondence by checking the consistency of the bidirectional deformation fields. The threshold τ is defined as follows:(3)$$\tau =\sum_{I_{m}}\frac{1}{N_{I_{m}}}\left(\delta \left(I_{m}\right)\right)+Const$$
where $N_{I_{m}}$ represents the total number of voxels in the moving image and Const represents a constant. We first calculate the average error of the moving image and set Const as 0.015 based on Tony’s work [26] to perform thresholding-based segmentation in order to obtain a logical non-correspondence area $Mask={⋁}_{voxel}^{I_{f}}\delta \left(I_{f}\right)>\tau$. For any voxels, if there is a significant violation of inverse consistency, the voxel reflects non-correspondence between the two images.

### 2.3. Stepwise Registration Network

The stepwise registration network, which deploys successive convolutional layers configured to capture deformations with high precision at each resolution scale, is key to our registration framework. As shown in Figure 3, our approach employs significantly larger convolution kernels in the initial level of the multi-level registration network. These expansive kernels enable the registration net to apprehend broad deformations across the image volume, providing the possibility to capture the overall spatial transformations and assimilate wide-ranging spatial information, which is crucial for the model to achieve the accurate initial comprehension of the paired registered images.

As the registration process advances into subsequent registration levels, the size of the convolution kernels is progressively reduced so that the network’s focus is gradually shifted to finer details, placing a growing emphasis on the relationships with adjacent voxels. By narrowing its scope, the network refines the deformation field, focusing on subtler discrepancies and aligning local structures. Each level of registration, with the exception of the initial level, is input with the upper level’s deformation field and a warped image is acquired via spatial transformation, as well as the attention map output from the corrected attention module, so that our network can refine the deformation field step by step.

### 2.4. Corrected Attention Module

To overcome the negative influence of non-correspondence voxels on registration, our model merges the meticulously designed stepwise attention with a principal network that introduces attention mechanisms to direct the network’s focus towards areas of substantial deformation. The structure of our model embodies the progression of the deformation fields evolving from coarse to fine. Each level of the registration network involves the generation of an attention map derived from the deformation field output from the upper level’s registration net. This allows the multi-level registration network to incorporate continuous attention to guide the registration of areas with large deformation.

A multi-level registration network forms multi-resolution image pyramids by using different downsampling schemes, which inevitably leads to weak textures or spatial aliasing, and so the deformation field output from the preceding level of registration net may be biased [25]. In clinical practice, the missing correspondence in the preoperative image should predominantly reside within the pathological area. More specifically, when designating a follow-up image to be the fixed image, the areas of non-correspondence in the preoperative image should, to the greatest extent possible, coincide with the pathological area of the preoperative image. To this end, we employ DMFNet [27] with provided pre-trained parameters to demarcate the pathological area within the preoperative image, consequently obtaining a segmentation mask for the pathological area of the preoperative image $M_{p}$. Similarly, when the preoperative image is used as the fixed image, the non-correspondence area in the follow-up image that has a deformation field should not be outside of the pathological area of the preoperative image. Thus, our stepwise attention, after being corrected, can be expressed as follows:(4)$$A_{cur}=CorrectedAttention\left(Upsample\left(\phi_{upper},\left(L_{cur},W_{cur},D_{cur}\right)\right)\right)⊕M_{p_{cur}}$$
where the current level’s attention $A_{cur}$ is derived from the upper level’s deformation field $\phi_{upper}$ that is upsampled to the size of the current level $L_{cur}\times W_{cur}\times D_{cur}$. The architecture of the corrected attention module is shown in Figure 4. In the following step, $A_{cur}$ is corrected by the current level’s preoperative pathological area $M_{p_{cur}}$.

### 2.5. Loss Function

The loss function of the proposed network is defined as follows:(5)$$L=L_{similarity}\left(I_{f},I_{w},\mathrm{Mask}\right)+\lambda L_{regularization}\left(\phi \right)$$
where $L_{similarity}\left(I_{f},I_{w},\mathrm{Mask}\right)$ is used to evaluate the similarity loss of registration quality between the fixed image $I_{f}$ and the warped image $I_{w}$, except for the non-correspondence area calculated by the forward–backward consistency constraint. This term aims to minimize the dissimilarity of corresponding areas between the two images. Specifically, we use normalized cross-correlation (NCC) as the similarity metric, given its effectiveness for image registration tasks [21]. The loss is calculated as follows:(6)$$L_{similarity}\left(I_{f},I_{w}\right)=-\mathrm{NCC}\left(I_{f},I_{w}\right)\times \left(1-Mask\right)$$
$L_{regularization}\left(\phi \right)$ is the regularization loss that ensures the smoothness of the deformation field ϕ. This is typically implemented as a penalty on the gradients of the deformation field. The regularization loss can be expressed as follows:(7)$$L_{regularization}\left(\phi \right)=\sum_{x}\left({∥\nabla \phi (x)∥}^{2}\right)$$
The parameter λ controls the balance between the similarity and regularization terms, ensuring that both the registration accuracy and the smoothness of the deformation field are appropriately weighted during model training.

## 3. Experiments and Results

### 3.1. Dataset

The experimental validation of our registration model was conducted on the Brain Tumor Sequence Registration (BraTS-Reg) public dataset, which is intended to establish a benchmark environment for deformable registration algorithms [28]. This dataset is a multi-institutional dataset that comprises 140 pairs of multi-modal magnetic resonance imaging (MRI) scans. All cases were diagnosed with glioma and clinically scanned using a multi-parametric MRI acquisition protocol.

Furthermore, all images in the dataset were pre-processed via skull-stripping, which involved extracting the brain tissue. The images were also resampled to a standardized size of 240 × 240 × 155 with a 1 mm^2^ spatial resolution. This uniformity in spatial dimensions ensures consistency across the dataset, facilitating the direct applicability and comparability of algorithms without introducing bias from pre-processing steps.

### 3.2. Experimental Details

The experiment was conducted using Python 3.9 as the programming language and PyTorch as the deep learning framework. In addition to PyTorch, we utilized several other Python libraries: NumPy for numerical operations, scikit-learn for five-fold cross-validation, and NiBabel for neuroimaging data processing. These libraries played a crucial role in facilitating the data handling and model implementation processes.

We resized all images to 160 × 160 × 80 to facilitate multi-level registration within the network, which is essential for optimizing the data for our specific model architecture and processing pipeline. We employed an NVIDIA RTX 4080 GPU for model training, running 1000 epochs with a batch size of 1, which was determined based on the available GPU memory. The parameter λ was set to 0.1. During model training, we used the Adam optimizer with an initial learning rate of 0.0001.

### 3.3. Evaluation Metrics

We utilized the target registration error (TRE) to evaluate our registration model. The BraTS-Reg dataset provides approximately 10 pairs of expertly annotated landmarks per patient, both in preoperative and follow-up MR scans. These landmarks, labeled to reflect the invariant anatomy despite the presence of a pathology, provide a ground-truth correspondence that is indispensable for the quantitative computation of the target registration error (TRE). The TRE is a widely used performance metric for landmark-based registration tasks; it measures the average Euclidean distance between the landmarks in the fixed image and the corresponding landmarks in the warped image. The TRE can be expressed as
(8)$$\mathrm{TRE}=\frac{\sum_{i=1}^{num}\sqrt{{\left(x_{f}-x_{w}\right)}^{2}+{\left(y_{f}-y_{w}\right)}^{2}+{\left(z_{f}-z_{w}\right)}^{2}}}{num}$$
where $\left(x_{f},y_{f},z_{f}\right)$ represent the coordinates of the landmarks in the fixed image, and $\left(x_{w},y_{w},z_{w}\right)$ represent the coordinates of the landmarks in the warped image. The term num represents the total number of landmarks.

### 3.4. Comparative Experiment

#### 3.4.1. Experiment Design

To evaluate the performance of our proposed method, comparisons with various state-of-the-art deformable registration algorithms were performed, including a widely used registration method implemented in the ANTs package SyN [15]; a deep-learning-based single-level registration method named VoxelMorph [21]; and a deep-learning-based multi-level registration method named DIRAC that had ranked first in another BraTs-Reg challenge [26], which was set as the baseline. We used NCC with the sampling radius set to 3 and multi-resolution optimization with three scales and 1000, 200, 50 iterations for SyN registration. We used the official implementation to build the model and chose NCC as the metric for similarity loss and the smoothness of the deformation field as the metric for regularization loss in VoxelMorph registration. Meanwhile, we set 0.0001 as the learning rate and used the Adam optimizer to train the model for 1000 epochs. We built the model and applied the same loss function as the official implementation, and used the same optimization method with 1300 epochs for DIRAC registration. We divided the dataset into a training set, a validation set, and a test set according to the ratio of 8:1:1 for all of the tested registration methods.

To further verify the generalization capability of our proposed method, we conducted a five-fold cross-validation experiment, comparing our method with the baseline. Firstly, the entire dataset was randomly divided into five equal folds. Each fold contained an equal number of cases, ensuring that the distribution of data was consistent across all folds. For each of the five-fold cross-validation tasks in this experiment, four of the folds were used as the training set, and the remaining fold was used as the validation set. This process was repeated five times, each time with a different fold serving as the validation set. This method ensured that each case in the dataset was used for both training and validation, providing a robust assessment of the model’s performance. After each fold’s training, the model’s performance was evaluated on the validation set, and the evaluation metrics were calculated for the TRE. The results from the five validation sets were then averaged to produce a final performance metric, offering a comprehensive assessment of the model’s generalization capability across different subsets of the dataset.

#### 3.4.2. Results and Analysis

Figure 5 shows some examples of the registration results obtained with the different methods, and the results visualized in the red box show that our method and the DIRAC method obtained registration results more similar to the fixed image. Table 1 reports the quantitative results for the test dataset compared with the state-of-the-art methods and shows that our model achieved the best registration accuracy over all compared methods. The traditional methods such as SyN and VoxelMorph had poorer performance in that they could not handle instances of considerably large deformation caused by areas of non-correspondence. Although the baseline (DIRAC) and our method both had better performance, our method improved the performance by 7.5% compared to DIRAC. Meanwhile, our model achieved the best registration performance in cases with large deformation (original TRE > 3 mm) among all of the methods.

Table 1 reports the quantitative evaluation of TRE values for the test dataset compared with the state-of-the-art methods. This table highlights the best-performing method in each case by showing the best TRE values in bold. Additionally, the mean and standard deviation (SD) of the TRE values are shown for each method in order to offer a comprehensive assessment of the registration performance achieved with each method. As seen in Table 1, our proposed method consistently outperformed the other methods across most cases, particularly in cases with significant deformation (original TRE > 3 mm). In cases with smaller amounts of deformation (original TRE < 3 mm), our method demonstrates performance levels comparable to DIRAC, showing the robustness of our approach under varying conditions of deformation.

In comparison to the traditional methods like SyN and VoxelMorph, which struggled with large deformations due to areas of non-correspondence, our method, along with DIRAC, shows a clear advantage. SyN and VoxelMorph exhibit significantly higher TRE values, indicating their limitations in handling registration task with non-correspondence areas. The mean TRE across all cases for our method is 1.85 mm, with a standard deviation of 0.83 mm, suggesting that our approach not only achieves the lowest mean TRE but also maintains consistent performance across different cases. This consistency is crucial for practical applications where reliable and accurate registration is essential. Moreover, our method demonstrates a notable improvement over DIRAC, particularly in cases with larger amounts of deformation. Specifically, our approach improves performance by 7.5% compared to DIRAC, highlighting the effectiveness of our method in the registration of preoperative and follow-up images of glioma patients. Figure 5 illustrates the visual results achieved with each of the tested methods, with those highlighted in the red box indicating that our method and DIRAC produce better registration results similar to the fixed image.

In summary, both the quantitative results and the visual results clearly demonstrate the superiority of our method in terms of both accuracy and consistency, making it a robust choice for image registration tasks, particularly in challenging cases involving large deformations.

As shown in Table 2, our method achieved better registration performance compared with the baseline. Figure 6 illustrates a comparison between the visual registration results achieved with DIRAC and those achieved with our proposed method. The areas in red represent non-correspondence areas calculated by both approaches, demonstrating that our model achieved more rational and interpretable visual results in terms of the non-correspondence areas. Additionally, we present the attention maps, which demonstrate that our corrected attention module effectively enhanced the registration network’s attention to areas with significant structural differences and pathological features. A comparative analysis was then conducted between the baseline and our proposed method, focusing on both the mean performance across folds and the variability to assess consistency and robustness. The averaged results from the five folds provide an overall evaluation of how well our method generalizes to unseen data compared to the baseline. Any significant differences in performance metrics between the methods were analyzed to highlight the strengths and weaknesses of each approach.

### 3.5. Ablation Experiments

In order to further demonstrate the effectiveness of the stepwise registration network and corrected attention module in the proposed model, we conducted ablation experiments. The results are shown in Table 3, and we can see that both the stepwise registration network and corrected attention module achieve effective improvements in performance. Additionally, we also performed ablation experiments based on the number of epochs and batch size. As shown in Figure 7, the results revealed that the best registration performance was achieved at 1000 epochs, while the batch size had a minimal impact on the results. Consequently, we selected a batch size of 1 for our model based on our GPU’s memory capacity.

## 4. Discussion

The registration of preoperative and follow-up MRI scans of glioma patients is challenging due to the existence of bidirectional non-correspondence areas and deformation areas large in volume. Our experimental validation demonstrated that our proposed method is able to (1) capture deformations accurately and refine the deformation field gradually, (2) introduce prior clinical knowledge into network training, and (3) perform better in registration tasks and achieve more rational visual results for non-correspondence areas.

Utilizing a bidirectional registration network is essential for capturing the complex transformations between preoperative and follow-up images, which both contain voxels lacking corresponding relationships. This approach mitigates the inherent limitations of traditional unidirectional registration by enabling an integrative perspective that accounts for deformations and non-correspondences in both imaging directions. Bidirectionality not only provides richer information for the refinement of the registration performance but also facilitates a more balanced and accurate assessment of the transformation consistency. Consequently, this strategy significantly improves the robustness of registration, which is required for precise clinical analyses and treatment planning.

By using the stepwise registration procedure, our model can generate accurate deformation fields at the initial level of the multi-level registration network and provide a precise deformation field for the subsequent levels of the multi-level registration network. This stepwise registration strategy provides a pivotal advantage over conventional multi-resolution registration methods in that it mitigates the risk of initial errors becoming magnified across the following levels of the registration. While traditional methods might compound the initial registration bias through successive resolution levels, our stepwise registration network avoids or reduces these biases initially and consequently prevents them from escalating as the registration network deepens. Our network delicately balances the capture of global and local deformation features, which results in more reliable and precise registration.

In order to reduce the adverse effects of non-correspondence areas during the process of multi-level registration, we must pay attention on the non-correspondence areas. Our method applies an attention map when generating a higher-resolution deformation field so that large deformation areas can be registered more accurately. The introduction of the prior knowledge of surgical resection in the pathological area makes the model constrained when calculated the missing correspondence area, and this can improve the treatment of missing correspondence areas in clinical practice.

Our method improves the registration performance between preoperative and follow-up MRI scans of glioma patients. However, some limitations exist in this study. Firstly, the reliance on a sparse set of landmarks, most of which were situated at a considerable distance from the tumor, hindered our ability to comprehensively assess the registration accuracy, particularly for areas close to the non-correspondence areas [29]. Furthermore, to the best of our knowledge, no established metric currently exists for evaluating the visual outcomes of medical image registration tasks involving missing correspondence. This is due to the lack of a ground truth for deformation fields or masks delineating non-correspondence areas. In response to these limitations, our future work will focus on enhancing the robustness and reliability of evaluations of medical image registration outcomes by generating synthetic data endowed with gold-standard deformation fields.

## 5. Conclusions

In this work, we have proposed a novel stepwise registration network complemented by a corrected attention module that is capable of learning large deformations with bidirectional non-correspondence areas, and it is completely unsupervised. This end-to-end framework is developed with the objective of mitigating or obviating the registration inaccuracy prompted by non-correspondence voxels, thereby enhancing the registration accuracy in normal brain tissue. By accurately capturing deformations and pinpointing areas of non-correspondence at the initial level of the registration network, our method effectively circumvents the biases in traditional multi-scale registration strategies. Additionally, our proposed corrected attention module is capable of refining the registration of non-correspondence areas and their surrounding areas. The experimental results demonstrate that our method leads to improved performance in both the TRE metric and the visualization of non-correspondence areas. In the future, we will focus on the generation of synthetic data annotated with gold-standard deformation field labels and the reasonable evaluation of non-correspondence areas. Through these advancements, we aspire to increase the utility and applicability of our technique in the realm of clinical practice.

## Figures and Tables

**Figure 1 bioengineering-11-00951-f001:**
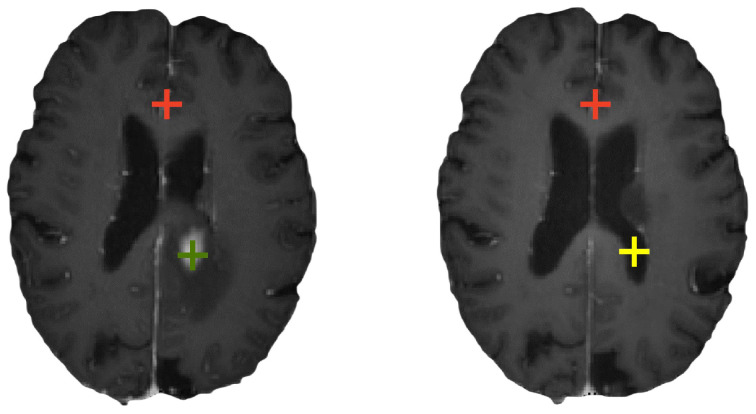
Different corresponding relationships in the brain MRI scans of glioma patients before and after surgery: red points have a corresponding relationship in both directions, the green point has no corresponding relationship from preoperative to follow-up, and the yellow point has no corresponding relationship from follow-up to preoperative.

**Figure 2 bioengineering-11-00951-f002:**
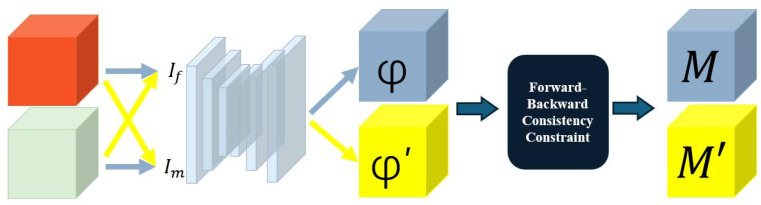
The bidirectional registration framework, wherein preoperative and follow-up images (denoted by the red and green blocks) are input as fixed and moving images, respectively. The non-correspondence areas can be located by calculating the forward–backward consistency constraint on the bidirectional deformation field.

**Figure 3 bioengineering-11-00951-f003:**
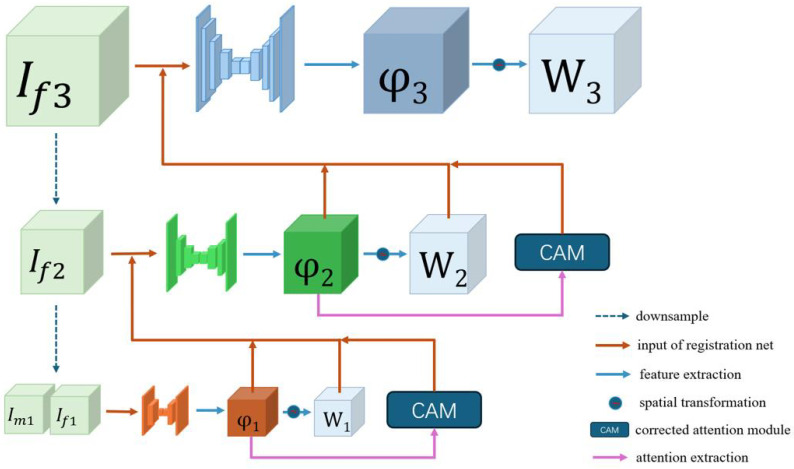
Overview of the proposed stepwise registration network, where $I_{f3},I_{f2},I_{f1},I_{m1}$ represent different resolutions in the interpolation of registered images; $\phi_{3},\phi_{2},\phi_{1}$ represent deformation fields output from each level of the registration network; and $W_{3},W_{2},W_{1}$ represent warped images of each level via spatial transformation. The different levels of the registration network have similar network architectures but different convolutional processes, so that our network can capture deformation features globally and then refine them locally.

**Figure 4 bioengineering-11-00951-f004:**
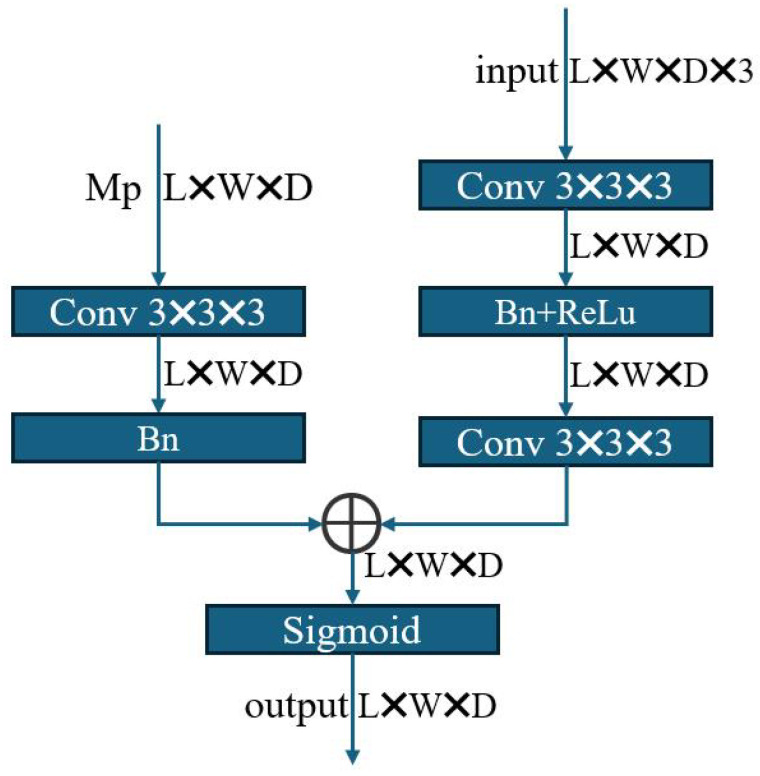
The designed corrected attention module, where the size of the input data changes with the transformation of the registration network, denoted as L×W×D and $\frac{L}{2}\times \frac{W}{2}\times \frac{D}{2}$, respectively.

**Figure 5 bioengineering-11-00951-f005:**
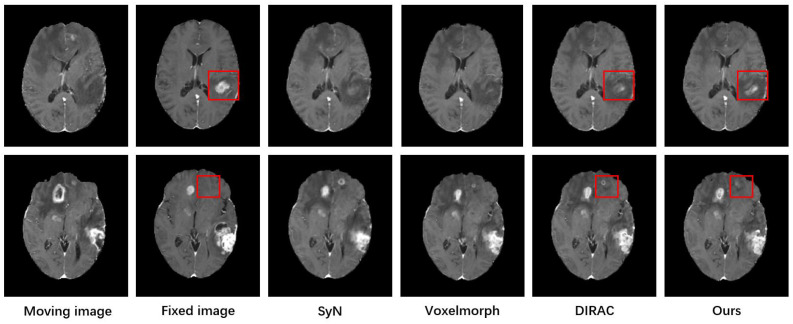
Registration results between preoperative and follow-up MRI scans with SyN, VoxelMorph, DIRAC, and our method. The registration results in the red box show that our method and DIRAC demonstrate better registration performance.

**Figure 6 bioengineering-11-00951-f006:**
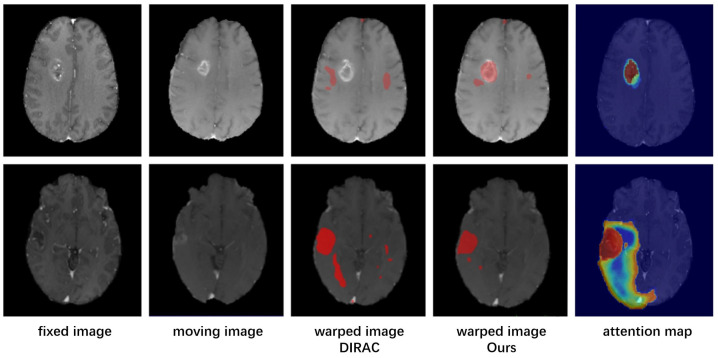
Visual results for the non-correspondence areas. Areas colored in red represent non-correspondence areas. In this paper, we compare our model with DIRAC, a method that ranked first in a previous BraTs-Reg challenge.

**Figure 7 bioengineering-11-00951-f007:**
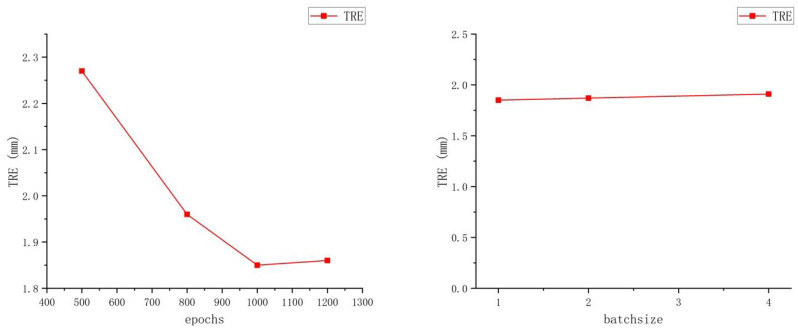
Quantitative evaluation of the impact of different numbers of epochs and batch sizes on registration performance. These results demonstrate that the optimal performance was achieved at 1000 epochs, while the batch size had a minimal impact on the results.

**Table 1 bioengineering-11-00951-t001:** Quantitative evaluation of the target registration error (TRE) in millimeters for different registration methods. This table presents the results across various cases, with the best-performing method in each row highlighted in bold. The mean and standard deviation of the TRE values are also shown for each method in order to provide an overall assessment of performance.

Origin TRE	Case	Origin	SyN	VoxelMorph	DIRAC	Ours
**<3 mm**	1	2.61	5.95	4.00	1.58	**1.57**
	2	1.61	14.77	2.56	**0.79**	1.03
	3	2.82	6.99	3.91	**1.26**	1.28
	4	1.62	1.70	2.45	1.15	**0.74**
	5	1.47	1.61	2.25	1.43	**1.40**
	6	2.54	5.63	4.19	**1.59**	1.76
**>3 mm**	7	3.51	8.41	5.37	1.93	**1.35**
	8	4.17	4.45	6.44	3.21	**2.79**
	9	3.77	6.60	9.21	2.89	**2.83**
	10	4.06	8.88	10.73	1.31	**1.24**
	11	6.56	14.83	10.68	2.44	**2.21**
	12	12.69	20.91	20.97	3.84	**3.79**
	13	19.65	42.18	26.81	2.51	**1.93**
	14	5.12	15.41	8.45	2.05	**2.05**
**Mean ± SD**		**5.16 ± 5.06**	**11.31 ± 10.05**	**8.43 ± 7.26**	**2.00 ± 0.88**	**1.85 ± 0.83**

**Table 2 bioengineering-11-00951-t002:** Quantitative evaluation of TRE (in mm) in the five-fold cross-validation experiment.

Fold	DIRAC	Ours
1	2.79	2.73
2	2.50	2.51
3	2.39	2.36
4	2.95	2.87
5	2.47	2.20
Average	2.62	2.53

**Table 3 bioengineering-11-00951-t003:** Quantitative evaluation of TRE (in mm) in the ablation experiments. These results demonstrate the effectiveness of both the stepwise registration network and the corrected attention module, with both contributing to improved performance.

Our Method		TRE Result (mm)
Stepwise Registration Network	Corrected Attention Module	
✓		1.95
	✓	1.92
✓	✓	1.85

## Data Availability

This study’s data are publicly available.

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
