# Peer review of "Stepwise Corrected Attention Registration Network for Preoperative and Follow-Up Magnetic Resonance Imaging of Glioma Patients"

_bioengineering, 2024, doi:10.3390/bioengineering11090951_

Round 1
Reviewer 1 Report
Comments and Suggestions for Authors
This is a very interesting manuscript, the authors propose a stepwise corrected attention method for registration between preoperative and follow-up MR images of glioma patients. The proposed method takes advantages of multi-level registration strategy capable to perform registration from coarse to fine resolution images. The authors suggest that the model can capture the complex and large deformations with high efficiency and accuracy. Furthermore a Corrected Attention module is introduced into the stepwise registration network and therefore the proposed model can continuously keep the attention on learning the deformation of large areas that. Finaly, the model was validated on public dataset and compared with other state-of-the-art deformable medical image registration methods.
The manuscript is well designed, written and presented. Here are a few comments that may help to improve it
1. Table1. Lines 1, 5 and 6 shouldn’t be also in gray background?
2. Please change the word bad “In order to reduce the bad effects” à “In order to reduce the adverse effects”
3. Providing the source code of your experiments on a public repository so that others may reproduce this work would a plus.
Reviewer 2 Report
Comments and Suggestions for Authors
The proposed paper deals with an interesting subject, but some minor and major omissions are highlighted in the following:
What is the role of constant Const from equation 3?
It is unclear if the dataset contains skull-striped images or if the segmentation method is proposed by the authors; please explain it.
In the section “2.2. Experimental details,” please add the usage libraries from Python. So in the same section, it is not mentioned if the images were resized.
Please insert in the section “2.4.2. Ablation Experiment” other ablation elements such as epochs number, and batch size.
In section “1.4. Corrected Attention Module,” the authors said that “we employ DMFNet to 183 demarcate the pathological area within the preoperative images, consequentially procuring 184 a segmentation mask for the preoperative pathological area Mp[27].” However, this CNN is not described; and it should be added to the abstract section.
Reviewer 3 Report
Comments and Suggestions for Authors
This paper proposes a stepwise corrected attention method for the registration of preoperative and follow-up brain MRI. Experiments were conducted on the BraTS-Reg dataset. The reviewer’s comments are as follows.
1) To demonstrate the effectiveness of the proposed attention module, it is required to visualize attention maps.
2) It is required to define the loss functions in Equation (5).
3) Experiments in Table 1 were conducted on the test set consisting of 14 images, while experiments in Table 2 were conducted based on 5-fold cross validation. What is the reason for utilizing both training-test split and 5-fold cross validation in the experiments?
4) In Fig. 6, the MRI in first column of the second row looks different to the MRI in the second column. Indicating the correspondence of MRIs will be helpful to understand the figure.
5) What is the difference in normal and bold versions of phi in Equation (2).
6) Typos and grammatical errors should be revised (e.g., \phi_{ohter} in line 143, Sigmod in Fig. 4, Lsimilarity in line 195 etc.) What is the meaning of “No.i” in lines 190-191?
Comments on the Quality of English LanguageThere are many typos and inconsistent notations. The authors should thoroughly revise the manuscript.
Round 2
Reviewer 2 Report
Comments and Suggestions for Authors
Your answers were punctual and the quality of the paper increased.
Author Response
Thank you again for all your excellent works on this paper and giving us valuable comments so that we can improve the clarity and quality of our manuscripts.
Reviewer 3 Report
Comments and Suggestions for Authors
The authors conducted major revision and most of the reviewer’s concerns are resolved. I have minor comments on the revised manuscript as follows.
1) In Equation (2), the notation of absolute value should be modified to the standard notation of L2-norm.
2) The “Mask” attribute is omitted in Equation (6).
3) In Figure 2, the authors should indicate the definitions of red and green blocks.
4) It is required to define the definition of V_voxel^{I_f} and \delta in line 151.
Comments on the Quality of English LanguageThe quality of English is fine.
